# Deep Learning to Predict the Cell Proliferation and Prognosis of Non-Small Cell Lung Cancer Based on FDG-PET/CT Images

**DOI:** 10.3390/diagnostics13193107

**Published:** 2023-09-30

**Authors:** Dehua Hu, Xiang Li, Chao Lin, Yonggang Wu, Hao Jiang

**Affiliations:** 1Department of Biomedical Informatics, School of Life Sciences, Central South University, Changsha 410013, China; 2Department of Nuclear Medicine & PET Imaging Center, The Second Xiangya Hospital of Central South University, Changsha 410011, China

**Keywords:** non-small cell lung cancer (NSCLC), Ki-67, PET/CT, deep learning

## Abstract

(1) Background: Cell proliferation (Ki-67) has important clinical value in the treatment and prognosis of non-small cell lung cancer (NSCLC). However, current detection methods for Ki-67 are invasive and can lead to incorrect results. This study aimed to explore a deep learning classification model for the prediction of Ki-67 and the prognosis of NSCLC based on FDG-PET/CT images. (2) Methods: The FDG-PET/CT scan results of 159 patients with NSCLC confirmed via pathology were analyzed retrospectively, and the prediction models for the Ki-67 expression level based on PET images, CT images and PET/CT combined images were constructed using Densenet201. Based on a Ki-67 high expression score (HES) obtained from the prediction model, the survival rate of patients with NSCLC was analyzed using Kaplan–Meier and univariate Cox regression. (3) Results: The statistical analysis showed that Ki-67 expression was significantly correlated with clinical features of NSCLC, including age, gender, differentiation state and histopathological type. After a comparison of the three models (i.e., the PET model, the CT model, and the FDG-PET/CT combined model), the combined model was found to have the greatest advantage in Ki-67 prediction in terms of AUC (0.891), accuracy (0.822), precision (0.776) and specificity (0.902). Meanwhile, our results indicated that HES was a risk factor for prognosis and could be used for the survival prediction of NSCLC patients. (4) Conclusions: The deep-learning-based FDG-PET/CT radiomics classifier provided a novel non-invasive strategy with which to evaluate the malignancy and prognosis of NSCLC.

## 1. Introduction

Lung cancer is the leading cause of cancer death worldwide, and it accounts for 18% of all cancer deaths [1]. Non-small cell lung cancer (NSCLC) is the main pathologic type of lung cancer, and it accounts for 85% of all lung cancer cases [2]. Many patients with NSCLC relapse after treatment, and treatment outcomes vary in patients with advanced stages of the disease [3]. Proliferating cell nuclear antigen (Ki-67) is a known malignant marker in cancers, including lung cancer, and it has been associated with tumor proliferation, invasion, metastasis and prognosis [4,5,6,7]. NSCLC patients with a high Ki-67 expression level have significantly lower progression-free survival (RFS) and overall survival (OS) [8,9]. This suggests that Ki-67 has important clinical value in the treatment and prognosis of NSCLC. Therefore, Ki-67 is recognized as a biological marker in NSCLC evaluation and has exciting potential as a prognostic factor of NSCLC [10,11,12].

Currently, Ki-67 expression level can only be decided through a postoperative specimen or needle biopsy [13]. However, these methods are not only invasive, but also may give incorrect results due to the spatial heterogeneity of the tumor, and may even lead to spread of the tumor. Therefore, a non-invasive method is urgently needed to evaluate the expression level of Ki-67. Fortunately, it has been reported that there is a strong correlation between radiomic features and the heterogeneity index at the cell level [14,15], and Ki-67 expression level has also been correlated with radiomic features [16,17]. Some studies have shown that these radiomic features can predict Ki-67 levels in lung cancer patients [18,19]. In NSCLC, CT and PET/CT are the preferred investigative tools for the diagnosis, staging and monitoring of treatment response. Gu Q et al. [13] used CT images to construct a random forest image omics classifier, a subjective imaging feature classifier and a combined classifier, and found that random forest had the best effect, while the combined classifier did not improve the prediction performance. Dong Y et al. [20] and Sun H et al. [21], respectively, developed a new nomograms based on imageology based on CT and dual-phase enhanced CT to predict the expression of Ki-67 in NSCLC patients. The results showed that the nomogram including the imaging score and related clinical factors was better than the imaging and clinical model. Other studies have also extracted the image features of PET/CT and evaluated the feasibility of predicting the expression level of Ki-67 from PET/CT images [22,23]. These studies have mainly been carried out based on the radiomics method of feature engineering, but the process is complicated and time-consuming.

In contrast, deep learning methods overcome these problems through a self-learning strategy and obtain better performances. At present, deep learning has achieved remarkable results in fundus images of patients with diabetic retinopathy, X-ray images of patients with COVID-19 and magnetic resonance images of brain tumors [24,25,26], indicating the feasibility of applying deep learning technology to medical image processing. Although a deep learning model usually needs a huge dataset, it has been confirmed that the image representation ability found in large-scale natural images can be applied to small medical sample images through transfer learning [27,28]. Meanwhile, to the best of our knowledge, there is no study that has used PET/CT images to construct a deep learning model to predict the Ki-67 expression levels and survival of NSCLC.

In this study, a convolutional neural network in a deep learning model was used to mine FDG-PET/CT image information, assess the Ki-67 expression level and build a risk model to predict the prognosis of NSCLC in a non-invasive manner.

## 2. Materials and Methods

### 2.1. Patients

We retrospectively collected the records of 381 patients with NSCLC from the Second Xiangya Hospital of Central South University, from August 2018 to December 2019. The inclusion criteria were as follows: (1) patients with disease confirmed by histopathological examination who had Ki-67 expression levels measured and (2) patients who underwent FDG-PET/CT examination within two weeks before any impaired operation. The exclusion criteria were as follows: (1) patients with other concurrent primary malignant tumors and (2) patients who received anti-tumor therapy before FDG-PET/CT examination.

After screening, a total of 159 eligible patients were eventually enrolled in the study, including 95 men and 64 women aged from 27 to 82. Among the 159 patients, 36 patients had lung squamous cell carcinoma and 123 patients had lung adenocarcinoma. In this study, information about survival time was obtained through the patients’ normative reviews or over the telephone.

### 2.2. PET/CT Image Acquisition

The PET/CT imaging instruments used were Biographym CTx PET/CT scanners (Siemens, Munich, Germany), the tracers were 18 F-FDG with radiochemical purity >95%, the synthesis equipment was the Siemens Eclipse RD accelerator and the Explora FDG 4 synthesis module was used. Before the tests, the patients fasted 6 h with fasting blood sugar <8.1 mmol/L, rested in a resting room for 15 min before injection, then took 18 F-FDG intravenously at a dose of 0.15 mCi/kg, and then rested calmly for 60 min for image capture. The scanning range was from the top of the skull to the upper part of the femur, and the scanning direction was from the pelvic cavity to the head. First, a CT scan (120 kV, 200 mA, scanning layer thickness of 3.75 mm) was performed, and then PET 3D acquisition. The acquisition speed was 2 min/bed, with a total from 6 to 7 beds. After acquisition, the image was rebuilt iteratively and the data were passed into the MMWP image post-processing workstation (Siemens, multi-modality workstation, Germany).

### 2.3. Ki-67 Expression Measurement

All specimens were surgically resected tumors obtained from needle biopsies (two or three reliable tumor tissue samples) to ensure that the Ki-67 detection represented the entire tumor. The EliVision Plus kit for instant immunohistochemistry (Fuzhou Maixin, Fuzhou, China) was used. The primary antibody was a mouse immunohistochemical monoclonal antibody against human Ki-67 antigen; refer to the kit instructions for the antigen–antibody reaction test. At 50 magnification, 1000 cells were randomly selected from each section and positive cells were counted. The Ki-67 index was calculated using the percentage of positive cells. According to the distribution of Ki-67 expression values in patients (the median value of Ki-67 expression was 25%) and supporting evidence from previous studies [8,22], the samples were classified into two groups: high Ki-67 expression (Ki-67 ≥ 25%) and low Ki-67 expression (Ki-67 < 25%).

### 2.4. Tumor Segmentation

PET/CT images of all patients were exported from the PACS system workstation in DICOM format, and then imported into the 3D Slicer software (version 4.11.20210226 R29738). To obtain the range of interest (ROI), tumors were first found in 3D view mode, and then the tumors of interest were drawn in 2D mode, and finally the segmentation image file in NRRD format was exported. Two diagnostic radiologists with 3 and 9 years of experience, respectively, completed the work by consensus; the whole process was carried out without knowing the patient’s pathological results.

### 2.5. Development of the Deep Learning Model

PET/CT images were separated into PET images and CT images. ROIs of PET images and CT images were adjusted to 64 × 64 pixels using cubic spline interpolation, and combined adjacent CT and PET were integrated into three-channel combined images in the order of CT, PET and CT (Figure 1). Then, the dataset was expanded via data enhancement; the average pixel value of the image was adjusted to 0 and the standard deviation was adjusted to 1 using Z-score normalization. After the above preprocessing, 1428 PET images, 2138 CT images and 1428 combined images were obtained. The data were divided into a training set and a test set according to the ratio of 7:3, and then the three groups of images in the training set were input into the Densent201 network for training [29]. We initialized the network parameters of DenseNet201 with uniform distribution to alleviate the problems of gradient disappearance, gradient explosion and slow convergence during the training of convolutional neural networks, and then trained 100 iterations at a 0.001 learning rate. To prevent memory overflow, we set the batch size to 32. After the training was completed, the test was carried out in 30% of the test set. The models can provide the probability of high expression of Ki-67 directly, without subsequent processing.

### 2.6. Statistical Analysis

The SPSS 26.0 (IBM, Armonk, NY, USA) statistical analysis software was used to perform the χ2 test or independent-samples t-test for the basic clinical data of patients. The difference was statistically significant at *p* < 0.05. A receiver operating characteristic curve (ROC) and decision curve analysis (DCA) method were used to evaluate the diagnostic performance of different classifiers. A Kaplan–Meier analysis and Cox regression analysis were used to analyze prognosis.

## 3. Results

### 3.1. Clinical Characteristics of Patients

Based on Ki-67 expression levels, patients were divided into two groups: 76 patients with high Ki-67 expression and 83 patients with low Ki-67 expression. The statistical analysis between age, weight, sex, pathological type and Ki-67 expression level showed that Ki-67 expression level was significantly associated with weight, sex and the pathological type of NSCLC (*p* < 0.05) (Table 1). High Ki-67 expression was commonly observed in male patient with greater weight and larger squamous cell carcinoma.

Follow-up data were collected from May 2019 to September 2021. The mean and median follow-up periods were 31.68 (95% CI, 30.70–32.66) and 33.00 (95% CI, 31.87–34.13) months, respectively. The survival rate was significantly higher in those patients with low Ki-67 expression (*p* < 0.01), which indicated that a high expression of Ki-67 was correlated with a poor prognosis of NSCLC (Figure 2). Images of patients with high and low expressions of Ki-67 are shown in Figure 3.

### 3.2. Performance of Deep Learning Models

Table 2 lists the AUC, accuracy, specificity and sensitivity of deep learning models based on PET, CT and FDG-PET/CT. The results indicated that the combined model had the best superior AUC, precision and specificity (0.968, 0.900 and 0.946, respectively), while the PET model showed superior results in accuracy and sensitivity (0.903 and 0.858) for the training test. Meanwhile, in the test set, the combined model had the highest score for AUC, accuracy, precision and specificity (0.891, 0.822, 0.776 and 0.902, respectively), while the PET model had the highest sensitivity (0.747). ROC curves of the three models are shown in Figure 4 below. Our results revealed that the combined model was superior for predicting Ki-67 expression.

### 3.3. Prognostic Study Based on Combined Model Parameters

The FDG-PET/CT combined model directly transformed the PET/CT images into a high expression score (HES), which indicated the probability that the model considers the patient to have a high level of Ki-67 expression. Generally, the “discrimination threshold” is set to 0.5, that is, when the score is greater than 0.5, it is judged to be a high expression; otherwise, it is judged to be a low expression. However, different threshold classifications result in different performance models. As shown in Figure 5, the sensitivity of the deep learning model reached 0.95 when the HES threshold was 0.39, and the Yoden index of deep learning model reached the maximum when the HES threshold was 0.58. In addition, when the HES threshold was 0.85, the specificity of the deep learning model reached 0.95.

The prediction model score of the Ki-67 expression level was used as the risk score for patients with NSCLC, and the 95% sensitivity threshold, 95% specificity threshold and maximum Yoden index threshold were used as the classification basis. The case samples were divided into four risk grades: low risk (risk score < 0.39), medium and low risk (0.39 ≤ risk score < 0.58), medium and high risk (0.58 ≤ risk score < 0.85) and high risk (risk score ≥ 0.85). As shown in Figure 6, overall survival in the test set was significantly stratified by the risk score (*p* < 0.0001), which revealed that the risk score was negatively associated with the survival of NSCLC patients. Meanwhile, the univariate Cox regression analysis further confirmed that HES was a risk factor for the prognosis of NSCLC (*p* < 0.01) (Table 3).

Additionally, we included four factors, HES, gender, age and SUVmax, into the prognostic model and obtained the nomogram (Figure 7). A nomogram is a visualization of a multifactor regression model. The value level of each factor is assigned according to the contribution degree of each factor to the outcome variable in the model, and then the scores are added to produce the total points. Finally, the probability of an individual’s outcome event is calculated through the functional transformation relationship between the total points and the probability of the outcome event. Figure 7 vividly demonstrates the impact of various variables on patient prognosis. Among them, HES is the main predictor of survival. A higher HES risk score, male gender, higher age and a higher SUVmax value indicate that a patient may have a lower survival rate. In Figure 8, the two-year survival rate predicted by the Cox model is close to the actual two-year survival rate. In general, our results demonstrated that the combined model was a promising tool to predict the proliferative capacity and prognosis of NSCLC.

## 4. Discussion

Precision medicine provides a good option for cancer management, which relies on validated biomarkers to classify patients’ possible disease risks, prognosis and/or treatment response. Therefore, the early and accurate diagnosis of lung cancer is especially important. Ki-67 is a biomarker with exciting potential in the diagnosis, treatment and prognosis of tumors. PET/CT combines the advantages of PET and CT images, and can accurately identify the location of a lung tumor, display the subtle structural changes of the tumor and also understand the functional metabolism of the lung tumor site. It is an important non-invasive diagnostic tool for non-small cell lung cancer. In addition, FDG-PET/CT can also reflect tumor heterogeneity [30]. Therefore, it is of great significance to predict the expression of tumor biomarker Ki-67 through FDG-PET/CT images, which provides a non-invasive method and may solve the problem of inaccurate biopsy caused by tumor heterogeneity.

Previous studies have shown that radiological features of images can non-invasively reflect underlying histopathological changes and the expression of some biomarkers, such as Ki-67 [18]. Moreover, there is a significant positive correlation between the FDG uptake rate in PET and the Ki-67 score [31,32,33,34]. Therefore, it is feasible to predict Ki-67 expression in NSCLC patients using PET/CT images, and this has been explored [22,23]. In this study, FDG-PET/CT images of NSCLC patients were used as the basis to construct a deep learning model to predict Ki-67 expression. The HES given by the deep learning model was incorporated into the clinical characteristics to construct a prognosis model. Both of them achieved good results.

In this study, we used a deep convolutional neural network to predict Ki-67 expression levels in NSCLC using non-invasive FDG-PET/CT images. The deep learning model demonstrated excellent performance in the test set (AUC = 0.891) and showed a significant correlation between high-dimensional image features of FDG-PET/CT and the Ki-67 expression level. Our study demonstrated that the FDG-PET/CT combined model had the best application potential, and therefore provides patients with a non-invasive way to assess Ki-67 expression levels, which is a good complement to tissue biopsies.

When we statistically analyzed the clinical data of two groups of patients with different Ki-67 expression levels, we found a correlation between clinical characteristics and Ki-67 expression levels, with high Ki-67 expression more commonly observed in patients with greater body weight, male gender, poorer differentiation and squamous cell carcinomas. Meanwhile, patients with high Ki-67 expression had a poor overall survival. Previous studies have demonstrated a significant association between clinical and radiological features and Ki-67 expression levels [16,18,35]. Clinical features such as age, sex, weight, smoking history, tumor stage and pathological subtypes have been used for predicting Ki-67 expression levels [36,37]. Our results were consistent with previous research and revealed that Ki-67 is a potential prognostic factor of NSCLC.

The expression level of Ki-67 reflects the rapid growth and malignant capacity of NSCLC and shows promise as an independent prognosticator of patient outcome [38,39]. Based on the combined model, we assessed the high expression probability of Ki-67 as a high expression score for NSCLC patients, and analyzed the correlation between high expression score and survival time using statistical analysis. Consistent with the actual situation, our results revealed that the high expression score was a risk factor for prognosis and significantly correlated with a poor survival of NSCLC patients. Therefore, among the models, the combined model had the highest superiority in Ki-67 expression and patient prognosis prediction.

Meanwhile, radiomic methods always use medical images to quantify tumor information at the macroscopic level and build the relationship between tumor images and Ki-67 expression level [13]. Although the radiology method has the advantages of small sample size and interpretability, it requires manual boundary labeling, which is tedious feature engineering. In contrast, the deep learning approach automatically learns the image features of Ki-67, avoiding the complex process of radiological methods. Therefore, we provided a promising prediction model for Ki-67 expression and the prognosis of NSCLC-based deep learning.

Despite the encouraging performance of the deep learning model, this study has several limitations. First, we only examined Asian patients, and Ki-67 expression levels may be affected by race. Second, the number of cases was too small, and the cases were not sufficiently representative. Third, no external validation was performed in this study. In future studies, the sample size should be expanded, and multi-center data should be collected for external verification to improve the performance and versatility of deep learning models.

In conclusion, the deep-learning-based FDG-PET/CT radiomics classifier could facilitate prediction of the expression level of Ki-67 and provided a novel strategy for assessing the cell proliferation and prognosis of NSCLC. Our study serves as a promising, non-invasive prognostic tool for clinical diagnosis of patients with lung cancer.

## 5. Conclusions

The deep-learning-based FDG-PET/CT radiomics classifier was shown to be a promising way to assess the proliferative capacity of NSCLC, and the prediction model we developed could effectively evaluate the survival of NSCLC patients. Therefore, our study provides a novel non-invasive strategy for evaluating the malignancy and prognosis of NSCLC.

## Figures and Tables

**Figure 1 diagnostics-13-03107-f001:**
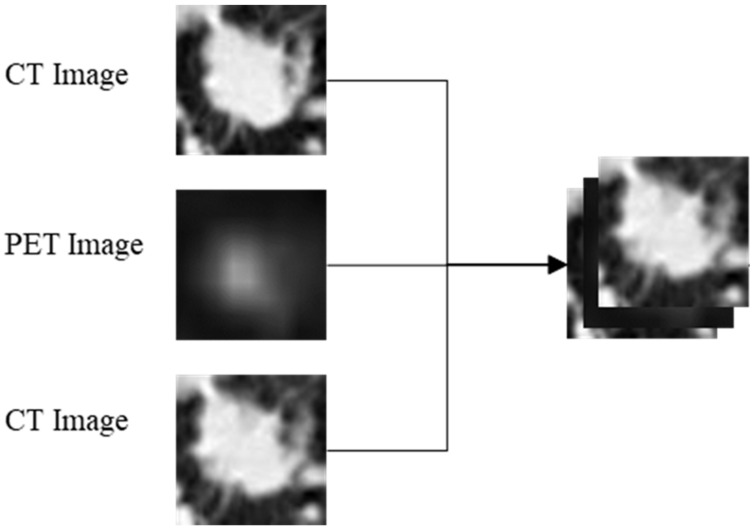
Combination of PET and CT images. The single-channel PET and CT images at corresponding positions were put into the new three-channel combination image in the order of CT, PET and CT.

**Figure 2 diagnostics-13-03107-f002:**
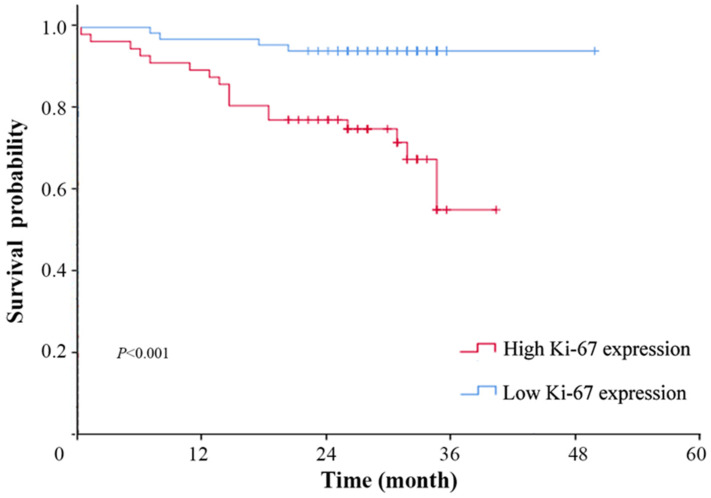
The survival rate of NSCLC patients with different Ki-67 expression levels. There was a significant difference in overall survival between the high Ki-67 expression group and the low Ki-67 expression group.

**Figure 3 diagnostics-13-03107-f003:**
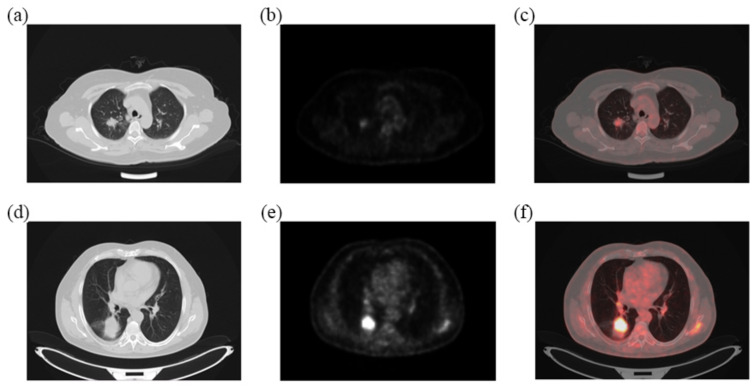
The FDG-PET/CT images of two patients: (**a**–**c**) A 57-year-old female patient with low Ki-67 expression; (**d**–**f**) a 62-year-old male patient with high Ki-67 expression. From left to right are CT images, PET images and fusion images.

**Figure 4 diagnostics-13-03107-f004:**
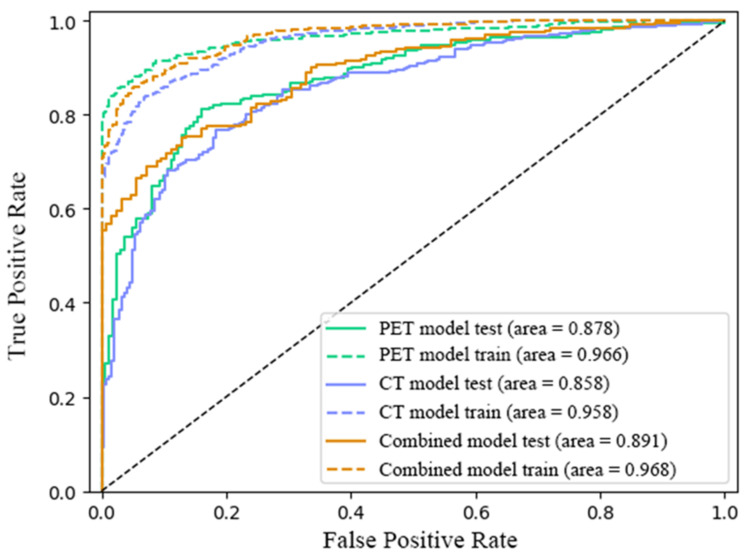
The ROC curves of the three models in training sets and test sets. The discrimination ability of the PET model, CT model and combined model in the training and testing set.

**Figure 5 diagnostics-13-03107-f005:**
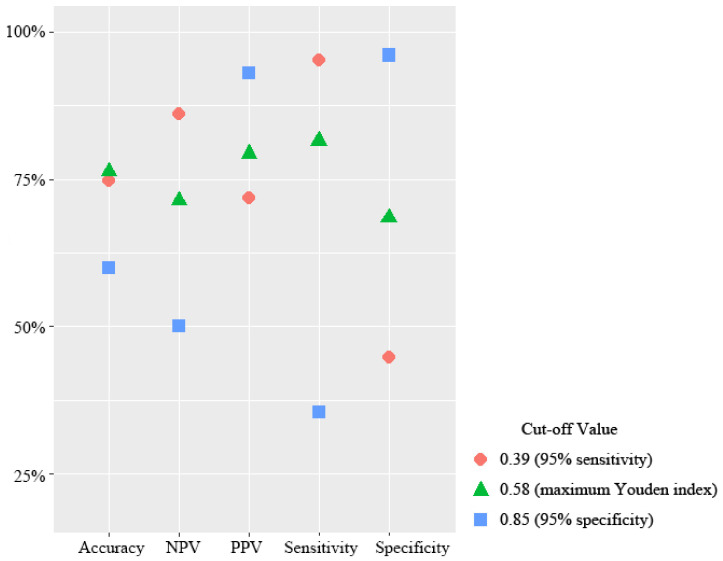
Predictive performance of the deep-learning signature in the determination of the Ki67 expression level. Charts show performance metrics of the deep-learning signature in the test set.

**Figure 6 diagnostics-13-03107-f006:**
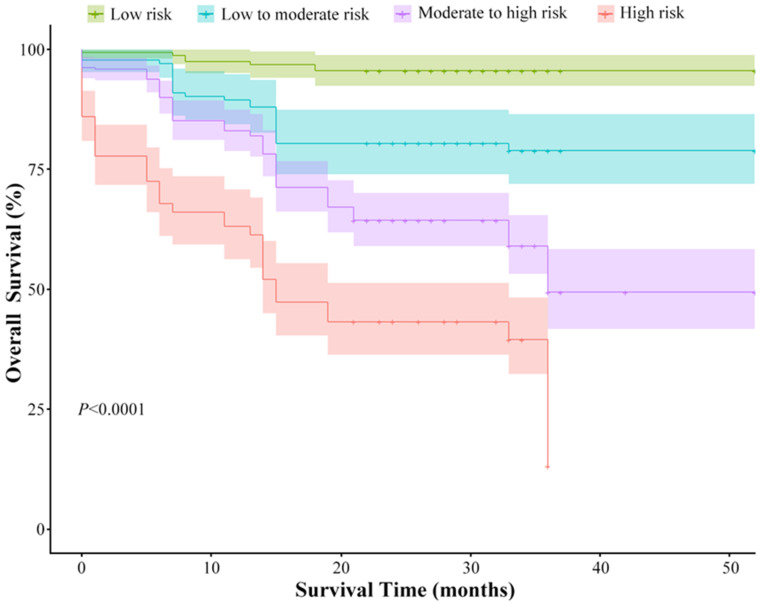
Prognostic value of the deep learning signature in non-small cell lung cancer. Survival curves of different risk groups defined by the deep-learning signature in the test set.

**Figure 7 diagnostics-13-03107-f007:**
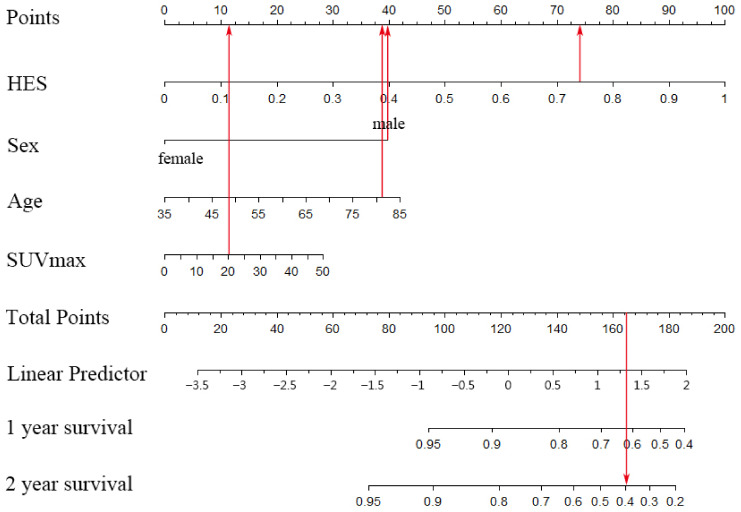
The nomogram in the test set. This nomogram includes four predictors: HES, age, sex and SUVmax. An 82-year-old male patient with an HES of 0.74 and SUVmax of 20.1 has a predicted probability of survival as shown by the red arrow. The total score of 164 predicted a 1-year survival probability of 0.62 and a 2-year survival probability of 0.40. The patient’s actual survival was 15 months.

**Figure 8 diagnostics-13-03107-f008:**
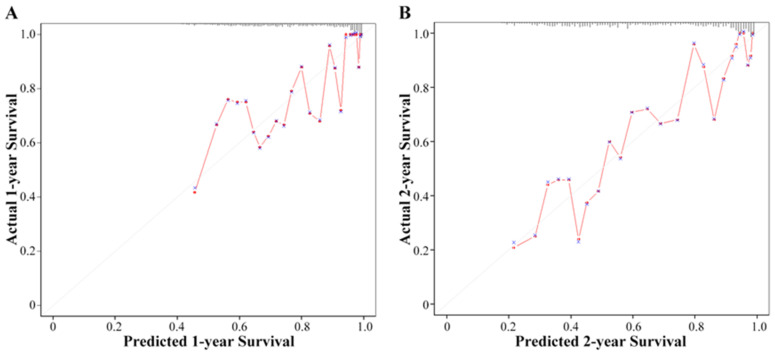
Calibration curves showed good calibration in (**A**) 1-year and (**B**) 2-year survival rate.

**Table 1 diagnostics-13-03107-t001:** Basic clinical data of the two groups of patients with non-small cell lung cancer.

Clinical Characters	High Ki-67 Expression (*n* = 76)	Low Ki-67 Expression (*n* = 83)	*p* Value
Age	60.83 ± 10.27	58.14 ± 9.54	0.093
Weight	62.03 ± 10.57	58.32 ± 10.26	0.027
Gender			<0.001
Male	64(40.3%)	31(19.5%)	
Female	12(7.5%)	52(32.7%)	
Differentiation state			<0.001
High	10(6.3%)	37(23.3%)	
Medium	19(11.9%)	35(22.0%)	
Low	47(29.6%)	11(6.9%)	
Histopathological type			<0.001
Adenocarcinoma	42(26.4%)	81(50.9%)	
Squamous cell carcinoma	34(21.4%)	2(1.3%)	

**Table 2 diagnostics-13-03107-t002:** Predictive performance of models.

Model	AUC	Accuracy	Precision	Specificity	Sensitivity	F1
PET Model						
Training set	0.966	0.903	0.884	0.930	0.858	0.870
Test set	0.878	0.804	0.747	0.840	0.747	0.747
CT Model						
Training set	0.958	0.872	0.819	0.882	0.856	0.837
Test set	0.858	0.794	0.754	0.853	0.702	0.727
Combined Model						
Training set	0.968	0.888	0.900	0.946	0.793	0.843
Test set	0.891	0.822	0.776	0.902	0.664	0.716

**Table 3 diagnostics-13-03107-t003:** Cox regression analysis.

Variable	Univariate Analysis		Multivariate Analysis	
	Regression Coefficient	*p* Value	Regression Coefficient	*p* Value
HES	3.986	<0.001	3.045	<0.001
Sex	1.818	<0.001	1.225	<0.001
SUVmax	0.042	<0.001	0.017	0.013
Age	0.037	<0.001	0.024	0.002

## Data Availability

The datasets generated during and analyzed during the current study are available from the corresponding author on reasonable request.

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
