# Peer review of "Deep Learning to Predict the Cell Proliferation and Prognosis of Non-Small Cell Lung Cancer Based on FDG-PET/CT Images"

_diagnostics, 2023, doi:10.3390/diagnostics13193107_

Round 1
Reviewer 1 Report
Hu et al. collected data from 159 patients in the hospital from August 2018 to December 2019 where patients were disease confirmed by histopathological examination and measured Ki-67 expression levels. And these patients underwent PET/CT within two weeks of histopathological examination. The data is analyzed through an existing Deep learning-based tool DenseNet-201 which enables the identification of cell proliferation antigen Ki-67 expression correlated with PET/CT images. The topic of work contributed to the recent advancement of health sciences. The manuscript may be accepted however, the following comment must be incorporated before further processing of publication.
1. The several technical terms should be similar throughout the manuscript, such as DenseNet-201, Ki-67, etc.
2. Mention the p-value in Figure 2 and the correct axis name of this figure.
3. The AUC values are seeming to not reflect the same as in Figure-4, please recheck all the values and answer the processor of calculations.
4. Figure 5 might be more compacted and enhance visibility. I think this is an important part of the recent work which is why needed to elaborate on the manuscript in detail.
5. How had you calculated the p-value for four survival curves in Fig-6?
6. Elaborate Fig. 7 in the text.
7. Manuscript has so many minor typographical and grammatical errors, that need to rectify.
Need to rectify grammatical and typological errors as mentioned in the author's comments.
Author Response
Response to Reviewer 1 Comments
Dear Editors and Reviewers,
Thank you very much for the feedback and comments on our manuscript,which are very valuable and helpful for us to revise and improve the paper. Accordingly, we made the changes and corrections as suggested, and we have used mdpi's language service to edit and review the article. We would appreciate very much if it can be accepted by your journal. The detailed responses are as following:
Point 1: The several technical terms should be similar throughout the manuscript, such as DenseNet-201, Ki-67, etc.
Response 1: Thank you for pointing this out. Thanks for your careful checks. We are sorry for our carelessness. Based on your comments, we have made the corrections to make these technical terms harmonized within the whole manuscript. And our amendments were marked in red in the revised paper.
Point 2: Mention the p-value in Figure 2 and the correct axis name of this figure.
Response 2: We appreciate your reminder, and we have updated Figure 2 on page 5. The legend information and axis name were corrected, and the p-values (p<0.001, p=0.00013) were added.
Point 3: The AUC values are seeming to not reflect the same as in Figure-4, please recheck all the values and answer the processor of calculations.
Response 3: We were really sorry for our careless mistakes. Thanks for your reminder. We have rechecked all the values in manuscript carefully and we have corrected the “AUC=0.968” into “AUC=0.891”.
Receiver Operating Characteristic curve (ROC), which can be used to compare the correlation performance of different classifiers, is plotted with the false positive rate (1-specificity) as the horizontal coordinate and the true positive rate (sensitivity) as the vertical coordinate. We got the probabilistic output of all samples through the deep learning model (we added some details to the deep learning model on page 4), and then plotted the ROC curve by changing the "discrimination threshold". The Area Under Curve (AUC) is defined as the area under the ROC curve, and a larger value means a better classifier. The above image rendering and the result settlement were implemented through python.
Point 4: Figure 5 might be more compacted and enhance visibility. I think this is an important part of the recent work which is why needed to elaborate on the manuscript in detail.
Response 4: Thanks for your advice. We have redrawn Figure 5 to make it compact and clear. We also added a more detailed explanation of Figure 5 on page 5, line 174. the specific contents are as follows:“The FDG-PET/CT combined model directly transformed the PET/CT images into a High Expression Score (HES), which indicated the probability that the model considers the patient to have a high level of Ki-67 expression. Generally, the "discrimination threshold" is set to 0.5, that is, when the score is greater than 0.5, it is judged a high expression, otherwise it is judged to be a low expression. However, different threshold classification results in different performance models. As shown in Figure 5, the sensitivity of the deep learning model reached 0.95 when the threshold was 0.39, and the Yoden index of deep learning model reached the maximum when the threshold was 0.58. In addition, when the threshold was 0.85, the specificity of the deep learning model reached 0.95.”
Point 5: How had you calculated the p-value for four survival curves in Fig-6?
Response 5: We used the “survdiff()” function in R to compare whether there was a difference between four survival curves, which compared whether there was a difference between the population. Both log-rank and Breslow test methods obtained p-values less than 0.0001. Then we used the “survminer” package's “pairwise_survdiff()” function in R to make multiple comparisons between survival curves. The test level was corrected using Benjamini and Hochberg FDR (BH), and the P-value between each group was also less than 0.0001.
Point 6: Elaborate Fig. 7 in the text.
Response 6: Thanks for your suggestion. We have detailed Figure 7 in the results section. The specific updates are as follows: “Additionally, we included four factors, HES, gender, age, and SUVmax, into the prognostic model and obtained the nomogram (Figure 7). A nomogram is a visualization of a multifactor regression model. The value level of each factor is assigned according to the contribution degree of each factor to the outcome variable in the model, and then the scores are added to produce the total points. Finally, the probability of an individual's outcome event is calculated through the functional transformation relationship between the total points and the probability of the outcome event. Figure 7 vividly demonstrates the impact of various variables on patient prognosis. Among them, HES is the main predictor of survival. Higher HES risk score, male gender, higher age and higher SUVmax value indicate that patients may have lower survival rate.”
Point 7: Manuscript has so many minor typographical and grammatical errors, that need to rectify.
Response 7: Thanks for pointing this out. We are sorry for our carelessness. We have checked the manuscript carefully and tried our best to polish the language. And we have used mdpi's language service to edit and review the article.
Again, we thank the editor and the reviewers for the very meaningful comments, and we believe the improvements made further strengthens the paper without changing the overall framework. All the major changes are marked in red for easy identification. We look forward to hearing your feedback.
Yours sincerely,
Hao Jiang
15 September, 2023
Reviewer 2 Report
The authors investigated the relationship between Ki-67 expression and prognosis of non-small cell lung cancer using deep learning. This paper lacks kindness and needs a general revision.
Specifically,
- P2, L80: “NSCLC patients treated”, what kind of “treated”?
- Any results of immunohistochemical analysis were not shown. Besides, the inaccuracy of needle biopsy due to heterogeneity of tumor did not seem to resolve.
- The median value of Ki-67 expression should be shown. It is unclear how to divide the two groups.
- The authors should present the detail of deep learning. What did you train on and what data did you test on? What is the basis for the CT, PET, and PET/CT?
- The calculation of high expression score is unclear.
- I do not understand why you are linking PET/CT images with Ki-67 expression. In addition, the image and the actual Ki-67 expression are not linking. Is it not possible to predict prognosis directly from PET/CT images?
- The authors discuss P11, L245-247, but in figure 2 the prognosis is better in the Ki-67 high expression group.
- P11, L251, an error occurred!
Author Response
Response to Reviewer 2 Comments
Dear Editors and Reviewers,
Thank you very much for the feedback and comments on our manuscript,which are very valuable and helpful for us to revise and improve the paper. Accordingly, we made the changes and corrections as suggested, and we have used mdpi's language service to edit and review the article. We would appreciate very much if it can be accepted by your journal. The detailed responses are as following:
Point 1: P2, L80: “NSCLC patients treated”, what kind of “treated”?
Response 1: Thank you for pointing this out. We updated the description in
manuscript. We collected data on patients not limited to a specific treatment, including surgery or other standard radiation and chemotherapy, and the largest number of patients were treated with surgery. All patients with pathologically confirmed NSCLC who had Ki-67 test results and PET/CT performed two weeks before surgery and did not meet the exclusion criteria were included in the study.
Point 2: Any results of immunohistochemical analysis were not shown. Besides, the inaccuracy of needle biopsy due to heterogeneity of tumor did not seem to resolve.
Response 2: Thank you for pointing this out. The immunohistochemical analysis index involved in this study was only "Ki-67", so we updated the expression on page 4 and line 104 of the manuscript and changed "immunohistochemical analysis" to " Ki-67 expression measurement ".
As for the second question, there is a strong correlation between the radiomic features and the heterogeneity index at the cellular level, and PET/CT can reflect the heterogeneity of the tumor. Therefore, the prediction of Ki-67 expression based on PET/CT images provided a non-invasive method to solve this problem. We added this point in the discussion section on page 12.
Point 3: The median value of Ki-67 expression should be shown. It is unclear how to divide the two groups.
Response 3: We appreciate your reminder, and we have added the median of Ki-67 expression in the section of Ki-67 expression measurement on page 3, line 113. At present, there is no clear classification criterion for the high or low expression of ki-67, so we used 25% as the classification criterion in this study based on the median expression of Ki-67 in 159 patients and the support of relevant studies (references[8,23]).
Point 4: The authors should present the detail of deep learning. What did you train on and what data did you test on? What is the basis for the CT, PET, and PET/CT?
Response 4: Thank you for your advice, We have filled in the details of deep learning in the Methods section on page 4. The details are as follows:
“PET/CT images were separated into PET images and CT images. ROIs of PET images and CT images were adjusted to 64×64 pixels using cubic spline interpolation, and combined adjacent CT and PET were integrated into three-channel combined images in the order of CT, PET and CT (Figure 1). Then, the data set was expanded via data enhancement, and the average pixel value of the image was adjusted to 0 and the standard deviation was adjusted to 1 using Z-Score normalization. After the above preprocessing, 1428 PET images, 2138 CT images and 1428 combined images were obtained. The data were divided into a training set and a test set according to the ratio of 7:3, and then the three groups of images in training set were input into the Densent201 network for training [29]. We initialized the network parameters of DenseNet201 with uniform distribution to alleviate the problems of gradient disappearance, gradient explosion and slow convergence during the training of convolutional neural networks, and then trained 100 iterations at a 0.001 learning rate. To prevent memory overflow, we set the batch size to 32. After the training was completed, the test was carried out in 30% of the test set.”
We used 70% of the data for training and 30% for testing.
CT stands for computerized X-ray Tomography. The principle is that according to the difference of X-ray absorption and transmittance of different tissues in the human body, a highly sensitive instrument is applied to measure the human body, and then the data obtained from the measurement are input into the electronic computer. After the data is processed by the electronic computer, the cross-section or three-dimensional image of the examined part of the human body can be taken to find tiny lesions in the body.
PET, the abbreviation of Positron Emission Tomography, is an imaging device that reflects the gene, molecule, metabolism and functional state of pathological changes. It uses positron-bearing radionuclides to label human compounds or metabolites as imaging agents (18F-FDG is the most widely used in clinic), which can be concentrated in tumor and other pathological tissues, thus showing a bright spot in the image and obtaining the information of human metabolic activity at the molecular level.
PET/CT combines the advantages of PET and CT images, which can not only provide accurate anatomical location of the lesion, display the subtle structural changes of the tumor, but also provide detailed molecular information of the function and metabolism of the lesion, and has the characteristics of sensitivity, accuracy, specificity and accurate location.
In the discussion section on page 12, we added the importance and feasibility of using PET/CT images to predict Ki-67 expression.
Point 5: The calculation of high expression score is unclear.
Response 5: high expression score (HES) is the probability that the deep learning model considers Ki-67 highly expressed in the patient, which is the direct output of the combined PET/CT model.
The PET/CT combined model is based on DenseNet201, which is a convolutional neural network with a depth of 201 layers. The DenseNet network structure mainly consists of three core structures, namely, DenseLayer (the most basic atomic unit of the model, which provides the initial feature extraction), DenseBlock (the basic unit of the model’s dense connection) and Transition (the transition unit among different dense connections for convolutional and pooling layers, which is used to integrate the learned features and reduce the size of the feature map). The model can be built through splicing and layer classification of the above structures. Each image generated a corresponding label according to the patient's group, with the low expression group labeled 0 and the high expression group labeled 1, as input to the deep learning model. After feature extraction learning, the output is the probability of predicting the image label as 1, that is, HES. Since the deep learning model is difficult to interpret, we regret that we are unable to give the specific process of HES calculation.
Point 6: I do not understand why you are linking PET/CT images with Ki-67 expression. In addition, the image and the actual Ki-67 expression are not linking. Is it not possible to predict prognosis directly from PET/CT images?
Response 6: The radiological features of the images can reflect the expression of Ki-67 non-invasive, and there is a significant positive correlation between the FDG uptake rate in PET and the Ki-67 score. Therefore, it is feasible to predict the expression of Ki-67 in NSCLC patients by PET/CT images. In the discussion section on page 12, line249-261, we added the importance and feasibility of using PET/CT to predict Ki-67 expression.
Prognosis can be predicted directly from PET/CT images. Radiological features can be extracted from images, and progression-free survival or overall survival can be assessed by Cox regression analysis after feature screening. However, we predicted the expression of Ki-67 from PET/CT images in this paper. Ki-67 as a biomarker, is related to tumor proliferation, invasion, metastasis, etc. Besides prognosis, it also plays a role in diagnosis and treatment. Our results have showed that Ki-67 has a strong ability to predict survival, so it could explain the ability of radiomic model to predict survival of lung cancer patients to a certain extent. Secondly, our prognostic model incorporated clinical features in addition to the information obtained from the images. Predicting prognosis directly from PET/CT images may not be as effective as models incorporating clinical features. Thank you for your question. We may be able to make a comparison between the two in the future research.
Point 7: The authors discuss P11, L245-247, but in figure 2 the prognosis is better in the Ki-67 high expression group.
Response 7: Thanks for your careful checks. The legend information in Figure 2 was incorrect. We are really sorry for our carelessness. We have updated Figure 2 and the legend information has been swapped on page 7.
Point 8: P11, L251, an error occurred!
Response 8: We were really sorry for the error when we inserted the citation, we have added the missing reference on page 13, line 294. (reference[13]).
Again, we thank the editor and the reviewers for the very meaningful comments, and we believe the improvements made further strengthens the paper without changing the overall framework. All the major changes are marked in red for easy identification. We look forward to hearing your feedback.
Yours sincerely,
Hao Jiang
15 September, 2023
Reviewer 3 Report
Hu et al. developed a deep learning tool based on combined FDG-PET and CT information to predict the tissue expression of a cell proliferation biomarker (Ki-67) in 159 patients with non-small lung cancer.
The rationale is sound and the results presented are convincing.
The following issues should be addressed to improve the quality of the report.
Major
English requires an in depth revision.
Line 89. Authors must report the instruments and the mean time of follow-up after diagnosis/ treatment of the 159 patients
Line 260. One additional major limitation of the study, to be included, is the lack of an external validatation.
Minor
The Authors seem to skip the possibility that PET in cancer patients can be used to reveal the distribution of several positron emitting radiotracers, while here they used the most common FDG-PET. Please insert FDG before PET throughout the title, manuscript, tables and legends.
Abstract
Line 21, Authors should detail which are the three models (FDG-PET, CT and combined FDG-PET/CT)
Introduction
Line 39-40. Please insert pertinent references.
Line 52 and 57. The term (traditional !) imageology is new to me and since the it sounds strange and unnecessary should be deleted. Thanks
Results
Line 141. It is not formally correct to insert in the Result section sentences as “Consistent with previous research”. Please delete.
Line 165. What is the “maximum idden index”?
Line 199. The fact that Fig.6 refers to stage I patients only is noteworthy and should be also indicated in the text.
Discussion
Line 222. Please insert the references concerning the statement that “Previous studies have demonstrated a significant association between clinical and radiological features and Ki-67 expression levels”.
Line 236. Please specify what the Authors mean for “high-dimensional PET/CT image”.
Line 251. A ref. is missing
Line 251. Please list here the advantages of the radiological approach.
Line 252. Please expand the meaning of the sentence “(the radiological features..) lacks the microenvironmental characteristics of the tumor”.
Conclusions
Line 268. Please substitute “established” with “developed”
An in depth revision of the english is mandatory
Author Response
Response to Reviewer 3 Comments
Dear Editors and Reviewers,
Thank you very much for the feedback and comments on our manuscript,which are very valuable and helpful for us to revise and improve the paper. Accordingly, we made the changes and corrections as suggested, and we have used mdpi's language service to edit and review the article. We would appreciate very much if it can be accepted by your journal. The detailed responses are as following:
Point 1: English requires an in depth revision.
Response 1: Thanks for your suggestion. We have tried our best to polish the language in the revised manuscript. And we have used mdpi's language service to edit and review the article.
Point 2: Line 89. Authors must report the instruments and the mean time of follow-up after diagnosis/ treatment of the 159 patients.
Response 2: Thanks for your reminder, we reported specific information about PET/CT imaging instruments on page 3, line 92. The median and mean follow-up times for patients were supplemented on page 5, line 156 in the results section, as shown below. “The mean and median follow-up periods were 31.68 (95% CI, 30.70–32.66) and 33.00 (95% CI, 31.87–34.13) months, respectively.”
Point 3: Line 260. One additional major limitation of the study, to be included, is the lack of an external validatation.
Response 3: Thanks for your valuable feedback. Despite our data set being limited in size and coming from a single center, we have employed rigorous methods and statistical analysis to ensure the reliability of our results. We performed data enhancement, dividing the training set and the test set in a 7:3 ratio, testing in 30% of the test set. Furthermore, our research results align with existing relevant studies and theories, further supporting the reliability of our conclusions. We added the limitations of the lack of external validation in discussion part on page 13, line 303, and encourage other researchers to actively conduct further studies using external validation datasets to validate our results. We appreciate your critical questions and suggestions, and we will continue to improve our research efforts.
“Third, no external validation was performed in this study. In future studies, the sample size should be expanded and multi-center data should be collected for external verification to improve the performance and versatility of deep learning models.”
Abstract
Point 4: The Authors seem to skip the possibility that PET in cancer patients can be used to reveal the distribution of several positron emitting radiotracers, while here they used the most common FDG-PET. Please insert FDG before PET throughout the title, manuscript, tables and legends.
Response 4: We sincerely thank you for careful reading. We have previously explained the type of PET/CT imaging instrument and the type of tracers used in the Materials and Methods section. As suggested by the reviewer, we have inserted FDG before PET throughout the title, manuscript, tables and legends. The changes were highlighted in the revised manuscript.
Point 5: Line 21, Authors should detail which are the three models (FDG-PET, CT and combined FDG-PET/CT)
Response 5: Thanks for pointing this out. We have added the specific names of the three models in the abstract section (p1, L22).
Point 6: Line 39-40. Please insert pertinent references.
Response 6: Thank you for your careful reading. After carefully examining the content of the article, we thought this sentence is not appropriate. In the revised manuscript, we have moved it to lines 48-50 and changed it to “Fortunately, there is a strong correlation between radiomic features and heterogeneity index at cell level, and Ki-67 expression level was also correlated with radiomic feature, and some studies have shown that these radiomic features can predict Ki-67 levels in lung cancer patients.”. And we have added corresponding references [14-19].
Point 7: Line 52 and 57. The term (traditional !) imageology is new to me and since the it sounds strange and unnecessary should be deleted. Thanks
Response 7: Thanks for your reminder. We have removed this term to make the article more fluent and logical.
Results
Point 8: Line 141. It is not formally correct to insert in the Result section sentences as “Consistent with previous research”. Please delete.
Response 8: Thank you for your reminder, we have deleted this sentence, and will pay more attention to the form of the article in the future.
Point 9: Line 165. What is the “maximum idden index”?
Response 9: We were really sorry for our careless mistakes. We have corrected the “idden index” into “Youden Index”. The Youden index represents the overall ability of the predictive method to detect genuine positive versus non-positive patients. The larger the value, the better the performance and the higher the authenticity of the prediction model. Line 165 indicates that the Youden index reaches its maximum value when HES threshold is 0.39.
Point 10: Line 199. The fact that Fig.6 refers to stage I patients only is noteworthy and should be also indicated in the text.
Response 10: We feel sorry for our carelessness. We mistakenly translated "early stage patients" to " patients in clinical Stage I." And we have removed ”clinical stage I”.
Discussion
Point 11: Line 222. Please insert the references concerning the statement that “Previous studies have demonstrated a significant association between clinical and radiological features and Ki-67 expression levels”.
Response 11: As suggested by the reviewer, we have added references to support this idea. (references[16,18,35])
Point 12: Line 236. Please specify what the Authors mean for “high-dimensional PET/CT image”.
Response 12: In fact, it was "high-dimensional PET/CT image features," which we changed to "high-dimensional image features of PET/CT" to avoid misunderstanding.
Point 13: Line 251. A ref. is missing.
Response 13: We feel Sorry for the error in our citation, and we have added the missing reference[13].
Point 14: Line 251. Please list here the advantages of the radiological approach.
Response 14: Thanks for your valuable comments. We added specific advantages of the radiological approach on page 13, line 294, such as the small sample size required and its interpretability.
Point 15: Line 252. Please expand the meaning of the sentence “(the radiological features..) lacks the microenvironmental characteristics of the tumor”.
Response 15: We have removed this sentence and updated the discussion section
Conclusions
Point 16: Line 268. Please substitute “established” with “developed”
Response 16: Thanks for your careful reading. Based on your comments, we have replaced the “established” with “developed” on page 14, line 315.
Again, we thank the editor and the reviewers for the very meaningful comments, and we believe the improvements made further strengthens the paper without changing the overall framework. All the major changes are marked in red for easy identification. We look forward to hearing your feedback.
Yours sincerely,
Hao Jiang
15 September, 2023
Reviewer 4 Report
The manuscript titled “Deep Learning to Predict Cell Proliferation and Prognosis of Non-small Cell Lung Cancer Based on PET/CT Images” by Hu and colleagues propose a novel non-invasive strategy to evaluate malignancy and prognosis of NSCLC. Overall the proposal is interesting, however, several comments below should be addressed to strengthen the study.
- The authors are encouraged to review the text carefully, such as line 12 “the current” repeated emergency.
- Is there a correlation between Ki-67 and SUVmax or tumor size? If not, how can PET/CT be used to determine KI67 expression?
- Line240 “The expression level of Ki-67 reflects the rapid growth and malignant capacity of NSCLC and shows promise as an independent prognosticator of patient outcome” Is there any literature to support this point? It has been reported that the expression of androgen receptor can affect the prognosis of Ki67, so when them both are expressed, Ki-67 is not associated with patient recurrence or survival. Please discuss it.
- The number of patients is small, and the follow-up time is not long. It would be better to improve this.
Not bad. The authors are encouraged to review the text carefully.
Author Response
Response to Reviewer 4 Comments
Dear Editors and Reviewers,
Thank you very much for the feedback and comments on our manuscript,which are very valuable and helpful for us to revise and improve the paper. Accordingly, we made the changes and corrections as suggested, and we have used mdpi's language service to edit and review the article. We would appreciate very much if it can be accepted by your journal. The detailed responses are as following:
Point 1: The authors are encouraged to review the text carefully, such as line 12 “the current” repeated emergency.
Response 1: We feel sorry for our carelessness. In our resubmitted manuscript, the mistakes are revised and we've deleted the redundant "the current" on page 2, line 12. Thanks for your correction.
Point 2: Is there a correlation between Ki-67 and SUVmax or tumor size? If not, how can PET/CT be used to determine KI67 expression?
Response 2: The correlation between Ki-67 and SUVmax, tumor size, or FDG uptake has been demonstrated in the references we cited [32-35]. In addition, we added the importance and feasibility of using PET/CT to predict Ki-67 expression in the discussion section on page 12, line249-261.
“PET/CT combines the advantages of PET and CT images, which can accurately locate the location of lung tumor, display the subtle structural changes of the tumor, and also understand the functional metabolism of the lung tumor site. It is an important noninvasive diagnostic tool for non-small cell lung cancer. In addition, FDG-PET/CT can also reflect tumor heterogeneity [31]. Therefore, it is of great significance to predict the expression of tumor biomarker Ki-67 through PET/CT images.
Previous studies have shown that radiological features of images can noninvasively reflect underlying histopathological changes and the expression of some biomarkers, such as Ki-67[18]. Moreover, there is a significant positive correlation between FDG uptake rate in PET and Ki-67 score [32-35]. Therefore, it is feasible to predict Ki-67 expression in NSCLC patients by PET/CT images, and this has been explored.”
Point 3: Line240 “The expression level of Ki-67 reflects the rapid growth and malignant capacity of NSCLC and shows promise as an independent prognosticator of patient outcome” Is there any literature to support this point? It has been reported that the expression of androgen receptor can affect the prognosis of Ki67, so when them both are expressed, Ki-67 is not associated with patient recurrence or survival. Please discuss it.
Response 3: Yes, we have supplemented the references in the corresponding (references [39,40]).
“It has been reported that the expression of androgen receptor can affect the prognosis of Ki67, so when them both are expressed, Ki-67 is not associated with patient recurrence or survival.”
Due to our carelessness, we misquoted this document. In fact, the conclusions of the literature are not consistent with our opinion, and the literature acknowledged that "Ki67 is independently associated with a higher risk of death." We have updated the references and removed this one reference.
Ki-67 high expression is more common in male, elderly, squamous cell carcinoma patients. Many studies have focused on the effects of estrogen to explain women's higher survival rates, and this article focused on the effects of androgens. This article argued that androgen receptor (AR) status seems to negate the association between high Ki-67 and poor outcomes, but his article explicitly mentioned that Ki-67 is independently associated with poorer survival and a higher risk of recurrence. And the conclusion of this paper was not supported by other literature. And our study did not measure androgen receptor (AR) status in patients and could not determine the effect of AR on Ki-67.
In the previous studies, radiomics showed a great ability to predict survival. However, the interpretability and rationality of the model have been questioned. Therefore, we used PET/CT to predict Ki-67 and further explained its role in prognosis. Our results showed that Ki-67 has a strong ability to predict survival, so the predictive ability of the radiomics model for survival of lung cancer patients can be explained to some extent.
Point 4: The number of patients is small, and the follow-up time is not long. It would be better to improve this.
Response 4: Thank you for your valuable feedback. Our dataset was limited in size, but we extended it with data enhancement methods to improve the reliability of our findings. Moreover, our model has achieved good performance, and the research results are consistent with the existing relevant studies and theories, which further supports our conclusion. Due to study time constraints, our follow-up time was not long, but we noticed that there were similar studies with similar follow-up time as ours [1,2]. Our model was effective in predicting two-year survival. We appreciate your critical questions and suggestions, and we will continue to improve our research efforts.
[1] Hindocha S, Charlton TG, Linton-Reid K, et al. A comparison of machine learning methods for predicting recurrence and death after curative-intent radiotherapy for non-small cell lung cancer: Development and validation of multivariable clinical prediction models. EBioMedicine. 2022;77:103911. doi:10.1016/j.ebiom.2022.103911
[2] Han EJ, Yang YJ, Park JC, Park SY, Choi WH, Kim SH. Prognostic value of early response assessment using 18F-FDG PET/CT in chemotherapy-treated patients with non-small-cell lung cancer. Nucl Med Commun. 2015;36(12):1187-1194. doi:10.1097/MNM.0000000000000382
Again, we thank the editor and the reviewers for the very meaningful comments, and we believe the improvements made further strengthens the paper without changing the overall framework. All the major changes are marked in red for easy identification. We look forward to hearing your feedback.
Yours sincerely,
Hao Jiang
15 September, 2023
Reviewer 5 Report
This study on using deep Learning to predict cell proliferation and prognosis in non-small cell lung cancer based on PET/CT images is certainly intriguing. While the idea of leveraging AI for prognosis prediction in NSCLC is a welcome advancement, it's essential to acknowledge that novelty might be a challenge in a field with substantial published research.
Additionally, relying solely on Ki-67 as a predictive marker for patient prognosis may not be sufficient, as NSCLC is a complex and heterogeneous disease. A more comprehensive approach that considers multiple biomarkers and clinical factors may yield more accurate prognostic predictions.
Nonetheless, this research is a significant step towards harnessing the potential of deep Learning in the field of oncology, and further exploration and refinement of the model could lead to more promising results in the future.
General is good.
Author Response
Response to Reviewer 5 Comments
Dear Editors and Reviewers,
Thank you very much for the feedback and comments on our manuscript,which are very valuable and helpful for us to revise and improve the paper. Accordingly, we made the changes and corrections as suggested, and we have used mdpi's language service to edit and review the article. We would appreciate very much if it can be accepted by your journal. The detailed responses are as following:
Point 1: This study on using deep Learning to predict cell proliferation and prognosis in non-small cell lung cancer based on PET/CT images is certainly intriguing. While the idea of leveraging AI for prognosis prediction in NSCLC is a welcome advancement, it's essential to acknowledge that novelty might be a challenge in a field with substantial published research.
Response 1: Thank you for pointing this out. Currently, there are a lot of studies on using artificial intelligence to predict NSCLC related biomarkers, but most of the studies use CT images as research objects. PET/CT integrates the advantages of PET and CT images, which may provide better model effects. We made predictions based on PET/CT images, and compared the performance of PET model, CT model and PET/CT combined model, proving the superiority of PET/CT combined model.
On the other hand, most of the current studies on PET/CT-based prognosis prediction of NSCLC are based on traditional radiomics methods. Deep learning can conduct end-to-end training, do not need to pay attention to the feature extraction process, and can extract deep abstract features that are difficult to extract by traditional methods, often with higher recognition accuracy. Therefore, this study was conducted based on deep learning, and good model performance was obtained even on the basis of small sample size, which provides evidence support for the application and promotion of deep learning in this field.
Point 2: Additionally, relying solely on Ki-67 as a predictive marker for patient prognosis may not be sufficient, as NSCLC is a complex and heterogeneous disease. A more comprehensive approach that considers multiple biomarkers and clinical factors may yield more accurate prognostic predictions.
Response 2: Thanks for your valuable feedback. NSCLC is a complex and heterogeneous disease, and other biomarkers besides Ki-67 are also related to prognosis. However, due to the limitation of research time and data, we only predicted the expression of Ki-67 in this study. In order to generate more accurate prognosis prediction, we combined clinical factors such as gender and age with HES to construct a prognostic model, and obtained better prediction results. We will continue to improve our research work by collecting more data and considering the prediction of other biomarkers, as well as combining multiple biomarkers and clinical factors to build more accurate prediction models.
Point 3: Nonetheless, this research is a significant step towards harnessing the potential of deep Learning in the field of oncology, and further exploration and refinement of the model could lead to more promising results in the future.
Response 3: Thank you very much for your comments and suggestions, we will continue to improve our research work in the future study. We will expand the sample size, include more influential factors, and further explore and improve the model to enhance its reliability and applicability.
Again, we thank the editor and the reviewers for the very meaningful comments, and we believe the improvements made further strengthens the paper without changing the overall framework. All the major changes are marked in red for easy identification. We look forward to hearing your feedback.
Yours sincerely,
Hao Jiang
15 September, 2023
Round 2
Reviewer 2 Report
I would like to thank for addressing all my comments. The manuscript is improved and I recommend for publication.